# 3D Model Identification of a Soft Robotic Neck

**Fernando Quevedo \*** , **Jorge Muñoz** , **Juan Alejandro Castano Pena** and **Concepción A. Monje**

RoboticsLab, University Carlos III of Madrid, Avenida Universidad 30, 28911 Madrid, Spain; jmyanezb@ing.uc3m.es (J.M.); jucastan@ing.uc3m.es (J.A.C.P.); cmonje@ing.uc3m.es (C.A.M.)
\* Correspondence: fquevedo@ing.uc3m.es

**Abstract:** Soft robotics is becoming an emerging solution to many of the problems in robotics, such as weight, cost and human interaction. In order to overcome such problems, bio-inspired designs have introduced new actuators, links and architectures. However, the complexity of the required models for control has increased dramatically and geometrical model approaches, widely used to model rigid dynamics, are not enough to model these new hardware types. In this paper, different linear and non-linear models will be used to model a soft neck consisting of a central soft link actuated by three motor-driven tendons. By combining the force on the different tendons, the neck is able to perform a motion similar to that of a human neck. In order to simplify the modeling, first a system input–output redefinition is proposed, considering the neck pitch and roll angles as outputs and the tendon lengths as inputs. Later, two identification strategies are selected and adapted to our case: set membership, a data-driven, nonlinear and non-parametric identification strategy which needs no input redefinition; and Recursive least-squares (RLS), a widely recognized identification technique. The first method offers the possibility of modeling complex dynamics without specific knowledge of its mathematical representation. The selection of this method was done considering its possible extension to more complex dynamics and the fact that its impact in soft robotics is yet to be studied according to the current literature. On the other hand, RLS shows the implication of using a parametric and linear identification in a nonlinear plant, and also helps to evaluate the degree of nonlinearity of the system by comparing the different performances. In addition to these methods, a neural network identification is used for comparison purposes. The obtained results validate the modeling approaches proposed.

**Keywords:** mathematical modeling of complex systems; non-linear models; soft robotics; soft robotic neck; tendon-driven actuators





## 1. Introduction

Soft robotics has been gaining importance in the robotics research field. The intrinsic compliance and adaptable properties of this hardware are pushing them into many areas. The purpose of these technologies is to overcome some of the problems found in the current robotic platforms. These include weight, cost, versatility and more importantly, safe human-to-robot interaction.

Different soft robotics technologies have emerged. These include pneumatic muscles with rigid links [1], pneumatic materials that deform according to their strain field [2], robots with fully inflatable links [3], fully inflatable robots [4], plant-based structures [5] and many other technologies [6,7]. In particular, we are interested in tendon-driven soft robots, a bio-inspired model scheme, as those in [8–10]. However, the kinematic models, unlike rigid ones, are not yet well-understood. Given the high non-linearity and physical characteristics, several assumptions and numeric simplifications are considered to actuate. Therefore, they are not as reliable and have lower versatility in comparison with their counterpart, thus limiting their impact on robotics [11]. These drawbacks are stopping soft robotics to enter fields such as industrial robotics or manufacturing. However, where

precision is not mandatory or humans might apply proper correction to achieve the desired goal, they have found a niche [12,13], such as rehabilitation and prosthetics.

In [8], the authors developed a mathematical model for the tendon-driven robotic arm. In particular, the authors approximated the elasticity of the tendons as a mass-less spring, given that their mass is many times smaller than the other parts (motors, gears, and loads). This enables them, on one side, to neglect coupling effects over different links, and also to assume rigid motions of the particular link to be modeled. Notice that even though the motion of the tendons is aligned with the arm motion, obtaining such a model is troublesome and requires further developments to increase the accuracy of the final result. To overcome the modeling problem for control purposes, the authors in [10] used reinforcement-learning to control the tendon-driven ACT hand synergies. This robotic hand has 24 motor-driven tendons that mimic the human hand biomechanics. Therefore, dynamic interaction with the hand skeleton results in redundant motions and other non-linear characteristics of the hardware that should be considered if a mathematical model is required. By using reinforcement learning, the authors are able to capture the desired dynamics over a set of motions that derive into a control strategy over the 24 tendons in a reduced state-space. However, as pointed out by [9], data-driven control algorithms on tendon-based robots or soft robotics have not yet been explored, and model-based control strategies are still preferred. Considering the previous statement, the authors in [9] analyzed an autonomous learning algorithm to obtain the model of a tendon-driven leg when different stiffness is used.

From the modeling perspective, [14] proposed a finite element model (FEM) for a glove with pneumatic bending actuators. The authors worked in a two-dimensional space, neglecting the dynamic energy from the model. In further research, a black-box model identification was given by [15] for a fluidic actuator. This allows the authors to introduce the shear deformation into the model, which in their previous work was not considered. The maximum parametric variations reached 36%, which was compensated by a back-stepping controller. In the present paper, black-box data-driven modeling will be used. Therefore, the overall dynamics of the given neck will be considered and modeling results will be compared with standard linear and non-linear modeling techniques, that is, Recursive Least Square and Neural Networks. This provides an overview of alternative modeling possibilities and their implications over a non-linear system as the tendon-driven neck.

In [16], a geometrical model and a two-dimensional FEM model for a soft fluidic actuator were studied. The geometrical model considered a uniform bending curvature of the link, while the FEM model showed a linear trend on the link behaviour. A new approach was proposed in [17] in order to obtain a pneumatic soft-arm 3D model, based on a constant link's bending curvature and neglecting the gravity or payload effects. Another geometrical approach for modeling a soft link is presented in [18]. In this case, the approached model neglects the effects of gravity or internal elastic forces. Regarding tendon-based robots, in the hand exoskeleton given in [19,20], each finger is considered as a three-link kinematic chain and all the joints are considered to be pure revolute. Friction and cable guide deformation were neglected. As shown in the cases above, geometric modeling requires several constraints and assumptions to reduce the model complexity, which allow the designers to cope with the complex system mechanics. Nonetheless, the black-box modeling approach will include the whole system dynamics, which led to the proposed methods in this work.

Modeling of soft robotic links is of particular interest, given the coupled dynamics that arise when actuating the robots. In this sense, different approaches are presented as well. Geometrical approaches expose their limitations when the modeling space increases. Therefore, sensor-based approaches are being used, although their results are limited to the sensor's range and capabilities. In [21], textile strain sensors are embedded into the robot structure to calculate the link deflection state and position. A similar approach is presented in [22,23]. In this last one, the authors describe the implementation of a soft hand where

each finger uses an elastic joint with an embedded piezo-electric transducer to sense the deflection of the joint. In this case, the curvature is given by the sensor but is assumed to be continuous through the link. A different sensor technique for embedded deflection modeling is presented in [24], where photo-sensing is used to determine the deflection angle of the link. A 3D modeling approach is given in [25], where embedded cameras for self-observing of the robot configuration are synchronized to obtain the final soft body (robot) shape through learning algorithms. The named references imply additional hardware but allow for rigid modeling based on sensor information. However, their effectiveness is limited to the sensor's accuracy and the validity of the hypothesis over the soft link, such as continuous link curvature. Furthermore, to obtain the mathematical model, it is still required to neglect certain dynamics, and in some cases, only the 2D motion of the soft link is considered. In this work, the soft link is aimed to move in 3D and it is not desired to add additional hardware to the system.

This work aims to characterize the 3D motion of a tendon-driven soft neck described in [26]. An improved version of the initial version was later proposed in [27]. This new design features a soft bendable spine that replaces the central spring. The replaced structure decreases the overall weight and increases the system robustness. An inclination sensor was also introduced on the top platform for extracting the current pitch and roll angles, and enables feedback control.

Although the central structure introduced an already non-linear behaviour, as seen in [27], the new material adds other non-linearity mechanics. Some previous works already tackle system identification. Firstly, in [26] for control design, the study was limited to actuators and ignored the dynamics of the link. The resulting theoretical model is outside the standard modeling methods. As a consequence, additional methodology is required to extract a simulation and control model for the platform. An initial identification exploration on 2D was presented in [28]. In that work, which this paper is a continuation of, we identified the soft link dynamics considering the actuators and the soft link. However, only the front inclination was considered, neglecting at that time the interactions that occur when all the soft neck degrees of freedom are used.

In this work, the proposed models are improved and extended to the entire robot motion range. Set membership and Recursive least-squares identification methods are used for modeling as in [28]. As the recursive least-squares method is only valid for linear plants, the non-linear behavior will not be captured. The selected methods do not need hardware modifications nor neglected dynamics. Therefore, physical effects, such as gravity, elasticity, and plasticity will be considered by the obtained 3D models when possible. These models are compared with a neural network model identification as ground comparison. As an important contribution, no modeling technique selected relies on local deformation sensors, and they do not require additional external hardware for possible neck control considered in the future.

The remaining parts of this paper are organized as follows. In Section 2, the platform to be identified is described. Sections 3 and 4 present the different methods used for identification. In Section 5 the experimental procedures are described and Section 6 shows the resulting models. Then, in Section 7, different tests are performed for validation and comparison purposes. Finally, in Section 8, the main conclusions are discussed.

## 2. Soft Neck Description

The mechanism that enables soft neck operation is the central soft link, which acts as a spine. It is made with bendable material and actuated with a parallel mechanism driven by cables, which produce a tilt in the upper platform. Figure 1 shows the soft neck prototype and its parts.

The neck is composed of a base, a mobile platform, a central soft link, tendons (cables), and motors, as shown in Figure 1. All parts were built using a 3D printer, including the soft link, which weighs 100 gr (excluding motors and hardware).

Combining the actions of the three actuators, any position or orientation can be reached inside the bounded space. As the robot workspace is three-dimensional, the final rotation can be defined using three Euler angles. The *Z*-axis rotation (yaw) is neglected since it cannot change due to the configuration of the link; therefore, two rotations around the *X*- and *Y*-axes are enough to fully define the robot position and orientation. System output will be defined through an *X*-axis rotation, roll($\phi$), and a *Y*-axis rotation, pitch ($\theta$), as shown in Figure 2.

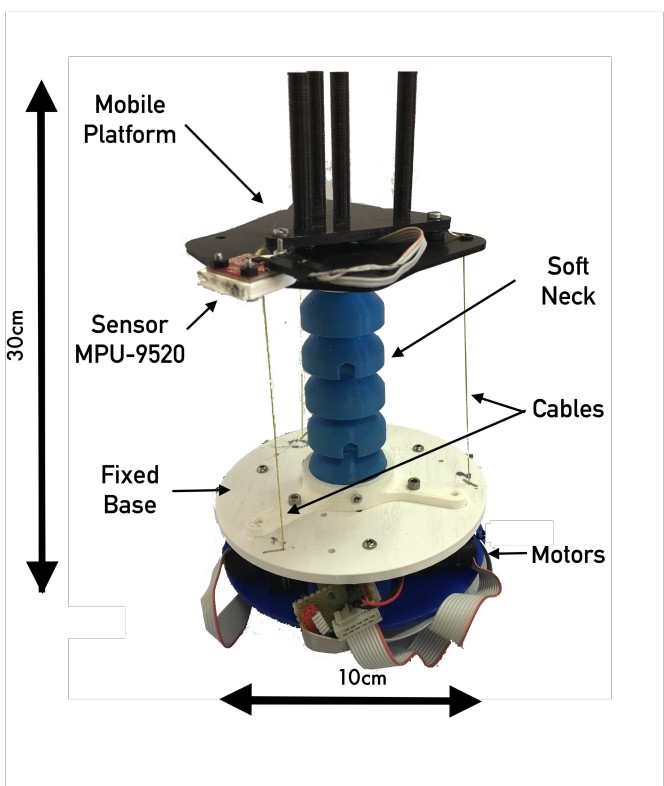

**Figure 1.** Soft neck platform.

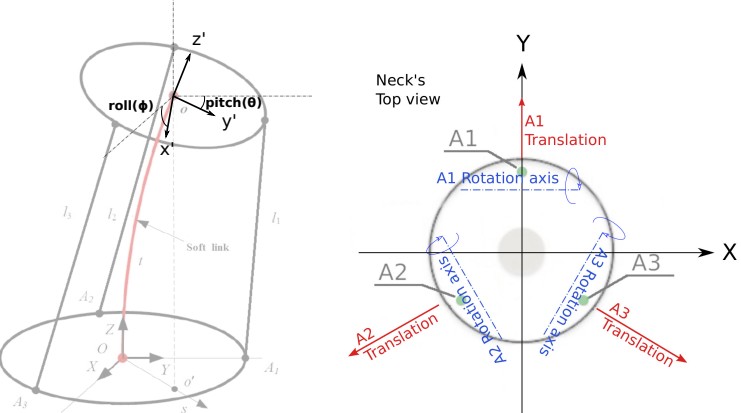

**Figure 2.** Soft neck kinematics. Orientation and inclination variables [27].

According to [29,30], the soft neck is a hyper-redundant robot. Therefore, the term degrees of freedom (DOF) is not applicable in the usual sense. Nevertheless, there is a connection between the three tendon lengths and the neck's final angular position.

The three tendon actuators are located at the base, each composed with a motor, gear, encoder, and a driver, with the characteristics shown in Table 1.

**Table 1.** Platform hardware specifications.

| | |
|---|---|
| **Driver** | Technosoft iPOS4808 MX-CAN; 400 W, 12–50 Volt, 8 Amp (intelligent motor driver) |
| **Motor** | Maxon RE 16-118739; graphite brushes, 48 Volt, 4 Watt |
| **Gear** | Maxon 134777 (24 : 1) |
| **Encoder** | Maxon mr201937 |

There exists low-level control managed by the motors' drivers, which satisfies all the platform's needs. For this reason, all system data are captured as an open-loop plan, with the actuator position and velocity as input and the inclinations roll and pitch as output.

*2.1. Geometric Simplification*

The described plant input and output variables are coupled, defining a Multiple Inputs Multiple Outputs (MIMO) system, which makes the system identification more difficult. Fortunately, there is a way to simplify the system, decoupling these variables, allowing to analyze its behavior through simpler Single-Input Single-Output (SISO) models. Using this scheme, the inverse kinematics described in [31] will not be necessary, making the model identification easier.

The angle combination produced by each actuator effect is a final rotation that can be defined or measured by two angles. A more detailed analysis of the robot's geometry will show the connection between the inputs and these final angle outputs. Since the inputs and outputs of the system are coupled in a MIMO system of three inputs and three outputs, it is desirable to rearrange the original input scheme using a linear combination of them. In this way, three new inputs will be obtained, having a direct action with respect to the outputs. Note that the aim is not to simplify the system, but to study the effects of the different inputs in the neck output variables. To simplify this operation, we will consider the system in the resting position. In this state, the single effect of the $A1$ actuator (reducing $l_1$) results in a rotation aligned with the $X$-axis of the base frame (see Figure 2). This results in an output angle directly related to the length of its tendon, and therefore, the motor position. Given that motor angles can be negative, we will consider the neck's rest position ($pitch = 0, roll = 0$) as the initial zero value for all tendons, resulting in positive values when tendon lengths increase, and negative otherwise. Considering just the first actuator with the index number 1, a possible equation describing pitch angle $\theta$ in $X$ is the following:

$$\theta_1 = f(P_1) \tag{1}$$

where $\theta_1$ is the angle contribution from the first actuator to the final pitch angle ($\theta$), $P_1$ is the actuator input position, and $f$ is a nonlinear function describing the relation between both. Although it is considered that just $P_1$ can change the angle $\theta_1$ as shown in Equation (1), given the neck's nonlinear nature, the other inputs may have a tiny effect on that angle, too, but they are considered too small and will be neglected in this case. The other actuators' effects ($\theta_2, \theta_3$) on the final angle $\theta$ are the following:

$$\theta_2 = \cos(\gamma_2)f(P_2) \tag{2}$$
$$\theta_3 = \cos(\gamma_3)f(P_3) \tag{3}$$

Given the proposed vertical robot setup, and using the same actuators, we can assume that the functions $f$ are similar. Nevertheless, a projection factor needs to be considered, which depends on the actuator's relative angle ($\gamma$). We can generalize the previous functions in the following equation:

$$\theta_i = \cos(\gamma_i)f(P_i) \tag{4}$$

Keeping in mind that we are not modeling the system but proposing an alternative input–output scheme, we can consider these input angles additive. Again, it is not a model simplification, but an input redefinition.

$$\theta = \theta_1 + \theta_2 + \theta_3 = \cos(\gamma_{11})f(P_1) + \cos(\gamma_{12})f(P_2) + \cos(\gamma_{13})f(P_3) \tag{5}$$

According to Figure 2, the three actuators are symmetrically arranged, and the angles are $\gamma_{11} = 0$ deg, $\gamma_{12} = 120$ deg and $\gamma_{13} = 240$ deg. Therefore, Equation (5) results in:

$$\theta = f(P_1) - 0.5f(P_2) - 0.5f(P_3) = f(P_1) - 0.5[f(P_2) + f(P_3)]. \tag{6}$$

This result shows how both $A_2$ and $A_3$ actuator effects on the angle *pitch* are divided by two, with an opposite direction to actuator $A_1$. This leads to the first result of this approach. The *pitch* angle is defined by the length difference, being positive when $P_1$ is larger than $0.5(P_2 + P_3)$, and negative otherwise. In the case of $P_1 = P_2 = P_3$, angle $\theta = 0$, leading to different robot compression states depending on the tendon lengths, form zero ($P_1 = P_2 = P_3 = 0$) to full compression ($P_1 = P_2 = P_3 = P_{max}$). This feature could be used to change the neck stiffness, although this is not discussed in this paper, where the soft link length is considered constant.

Now, roll angle ($\phi$) is defined as the rotation around the $Y$-axis. Using the previous reasoning but projecting in the $Y$-axis (using $\sin(\gamma)$):

$$\phi = \phi_1 + \phi_2 + \phi_3 = \sin(\gamma_1)f(P_1) + \sin(\gamma_2)f(P_2) + \sin(\gamma_3)f(P_3) \tag{7}$$

In the case of $\gamma_1 = 0$ deg, $\gamma_2 = 120$ deg and $\gamma_3 = 240$ deg, Equation (7) results in:

$$\phi = 0.866f(P_2) - 0.866f(P_3) = 0.866[f(P_2) - f(P_3)] \tag{8}$$

Note that in this case, the value of the $\phi$ angle just depends on the difference between $P_2$ and $P_3$, and that the $A_1$ actuator has no effect. Again, the angle just depends on their difference, and the compression is an average function of the tendon lengths. For the case $P_1 = P_2 = P_3$, angle $\phi = 0$, leading to the same previous result regarding soft link compression. Additionally, note that $\theta$ and $\phi$ angles depend on the tendon length difference, and the compression ($\delta$) depends on the tendon lengths' average. Based on this, we can define the new input variables $\theta_i$, $\phi_i$ and $\delta_i$ as a linear combination of the motor position inputs.

Using the results from Equations (6) and (8), and considering the link compression input as the motor positions' average, the following input redefinition is proposed:

$$\theta_i = P_1 - 0.5(P_2 + P_3) \tag{9}$$

$$\phi_i = 0.866(P_2 - P_3) \tag{10}$$

$$\delta_i = \frac{P_1 + P_2 + P_3}{3} \tag{11}$$

Using this input redefinition, we can decouple and simplify the system considering $\phi_i$ as an input, which provides a change exclusively in the $\phi$ output angle. Therefore, a nonlinear single-input single-output (SISO) system can be defined, having $\phi_i$ inputs and $\phi$ outputs. Likewise, $\theta_i$ and $\delta_i$ inputs will affect only the output values of $\theta$ and $\delta$, respectively, defining another two SISO systems.

Based on this, the soft neck can be modeled as three decoupled SISO systems. The transfer functions $G_\theta$, $G_\phi$, and $G_\delta$ will model the actual outputs ($\theta,\phi,\delta$) as a function of the new inputs ($\theta_i,\phi_i,\delta_i$), defined by Equations (9)–(11). Given the simplifications we have considered, the real behavior will be different in several aspects, like showing interference between actuators and a nonlinear plant response. These effects will be discussed in the Experiments section.

Two different system identification methods are used. First, the set membership method as described in [32] is used for nonlinear identification, and second, recursive least-squares (RLS), as described in [33], is applied for different tilt configurations, which will result in a linear system for each RLS identification performed. The evolution of these systems, according to the inclination, will be studied.

In the case of RLS system identification, the new redefined inputs ($\theta_i$, $\phi_i$ and $l_i$) were considered instead of motor position inputs. Note that these are just the input redefinition, and the output angles still depend on the system dynamics. Although $f$ functions are unknown, they are considered within the resulting models, although the nonlinear part may be neglected depending on the identification method.

### 3. Set Membership Non-Linear Identification

This section briefly describes the Non-Linear Set Membership (NLSM) identification method proposed in [32].

Consider a system that has a Nonlinear AutoRegressive with eXogenous input (NARX) structure, as

$$y(k) = f_o(\boldsymbol{\omega}(k)) + e(k) \tag{12}$$

where $\omega(k)$ is the system's regressor formed by past samples of the system inputs $u1, u2$ and the output $y1, y2$, as:

$$\boldsymbol{\omega}(k) = [y_i(k-1), \ldots, y_i(k-n_y), u_i(k-1), \ldots, u_i(k-n_u)]' \tag{13}$$

$$\boldsymbol{\omega}(k) \in W \subseteq \mathcal{R}^n, n = \sum_i n_{y_i} + n_{u_i} \tag{14}$$

where $e(k)$ represents the measurement noise and $W$ is the function domain.

The NARX regressor is widely used in system identification considering its capacity of representing nonlinear dynamics and developing estimation algorithms which are computationally cost-efficient.

If $f_o$ is unknown, but a set of measurements of $y_i(k)$ and $\boldsymbol{\omega}(k)$ are available for $k = 1, ..., N$ and considering that the noise magnitude is bounded by $\epsilon$:

$$|e(k)| \leq \epsilon \tag{15}$$

and no statistical assumption on its behavior is made. The goal is to estimate $\hat{f}$ of $f_o$, where $\hat{f}$ is the estimation of $f$.

Even though $f$ is unknown, the following information is available:

$$f_o \in \mathcal{F} \doteq \left\{ f \in C^1(W) : \|f'(\boldsymbol{\omega})\| \leq \gamma, \forall \boldsymbol{\omega} \in W \right\} \tag{16}$$

where $f'(\boldsymbol{\omega})$ denotes the gradient of $f(\boldsymbol{\omega})$ and $\|x\|$ is the Euclidean norm. Therefore, we assume that the identified system is continuous on its first derivative and has maximum growth of $\gamma$ for all the regressors applied to the function of interest.

On the other hand, if there is a Feasible System Set (FSS), which is the set of all systems in the space $\mathcal{F}$ which satisfies the following conditions:

$$FSS \doteq \left\{ \begin{array}{c} f \in \mathcal{F} : |y(k) - f(\boldsymbol{\omega}(k))| \leq \epsilon, \\ and \\ f \in \mathcal{F} : \frac{y(k) - y(k+1)}{\delta_T} \leq \gamma \end{array} \right\}, k = 1, 2, \ldots, N \tag{17}$$

therefore, there always exists a non-empty $FSS$ and $f_o \in FSS$ when both assumptions on $f_o$ and $e$ are true. Then, if we guarantee the validity of the conditions $\gamma$ and $\epsilon$ over a set of measurements generated by the system to be identified, we will find a $FSS \neq \varnothing$. In [32], the procedure to guarantee conditions $\gamma$ and $\epsilon$ over a data set is presented. For the following sections, prior assumptions are considered to be true.

Given that the aim of the model is to find the output generated by the system for a new input, it is necessary to distinguish the identification data set, $k$, and the new inputs $x$. Hence, for a given input $x \in W$, the optimal NLSM estimate of $f_o(x)$ is:

$$f_c(x) \doteq \frac{f_u(x) + f_l(x)}{2} \tag{18}$$

where:

$$f_u(x) = min_{1 < k < N} y(k) + \epsilon + \gamma \|x - \boldsymbol{\omega}(k)\| \tag{19}$$
$$f_l(x) = max_{1 < k < N} y(k) - \epsilon - \gamma \|x - \boldsymbol{\omega}(k)\| \tag{20}$$

As presented in [32],

- $f_u(x)$ and $f_l(x)$ are optimal upper and lower bounds for $f_o(x)$, respectively.
- $f_u(x)$ and $f_l(x)$ are Lipschitz-continuous on $W$; therefore, they belong to the FSS.
- $f_c(x)$ is an optimal approximation of $f_o(x)$ for any $L_p(W)$ norm, with $p \in [1, \infty]$, with an optimality criterion as:

$$f_{opt} = \arg \inf_{\hat{f}} \sup_{f \in FSS} \left\| f - \hat{f} \right\|_p$$

The NLSM algorithm produces a non-linear, non-parametric model which is embedded on the data set. That is, there is no explicit equation that represents the input–output or physical variables relation.

For a new regressor value $x \in \mathcal{R}^n$, the NLSM model output $f_c(x)$ is evaluated through Algorithm 1.

---

**Algorithm 1:** Set membership algorithm.

$F_{NLSM}(x)$
Set $f_u(x) = +\infty$
Set $f_l(x) = -\infty$
**for** $k = 1$ *to* $N$ **do**
    Calculate the distance between $x$ and $\boldsymbol{\omega}(k)$ as
    *Distance*$(k) = \|x - \boldsymbol{\omega}(k)\|$.
    *Obtain the upper bound on $f_o(x)$ guaranteed by $\boldsymbol{\omega}(k)$ as the projection*
    $P_u(k) = y(k) + \epsilon + \gamma * Distance(k)$.
    *Obtain the lower bound on $f_o(x)$ guaranteed by $\boldsymbol{\omega}(k)$ as the projection*
    $P_l(k) = y(k) - \epsilon - \gamma * Distance(k)$.
    *Choose the lowest upper bound*
    **if** $P_u(k) \leq f_u(x)$ **then**
       | $f_u(x) = y(k) + \epsilon + \gamma \|x - \boldsymbol{\omega}(k)\| = P_u(k)$
    **end**
    *Choose the highest lower bound*
    **if** $P_l(k) \geq f_l(x)$ **then**
       | $f_l(x) = y(k) - \epsilon - \gamma \|x - \boldsymbol{\omega}(k)\| = P_l(k)$
    **end**
**end**
Calculate the estimation
$f_c(x) = \frac{f_u(x) + f_l(x)}{2}$
**return** $f_c(x)$

---

In order to obtain the FSS, as described in [32,34], it is possible to execute the Algorithm 1 over the identification data set, updating the variable $\gamma$ whenever the positive or negative projections $f_u(i)$, $f_l(i)$, over each data point $\boldsymbol{\omega}(i) \in \boldsymbol{\omega}(k) \forall k \neq i$ produces a greater, $f_u(i) < y(i)$, or lower, $f_l(i) > y(i)$, value.

### 3.1. Non-Linear Set Membership Data Set Generation

In our specific problem, as described in Section 2, we have three motors that drive three tendons to actuate over the soft link and provide the desired pitch-roll motion. As described in Algorithm 1, to provide an estimation using Set Membership, we need to construct the FSS for a defined regressor that contains enough information so that all the identified system behaviors are contained in the FSS.

For this purpose, we define our regressor empirically since there are no standard methodologies to do so. Therefore, for simplicity, we run several system identifications using a neural networks MATLAB toolbox providing non-linear data-driven models for different regressor sizes. Once the NN is trained, we assume that the best neural network model uses the most informative regressor and corresponds to the original regressor selection. Later, to improve the computational time, the regressor is reduced by running different estimations modifying the number of elements in the regressor. In this way, less operations are required for each of the estimated data [35]. The chosen regressor is:

$$
\begin{aligned}
\boldsymbol{\omega}(k) \quad = \quad & [y(k-1), u_1(k-2), u_1(k-3), \\
& u_2(k-2), u_2(k-3), \\
& u3(k-2), u3(k-3), \\
& M_1(k-3), \\
& M_2(k-3), M_3(k-3)]
\end{aligned}
\tag{21}
$$

where $u_i(k)$ is the desired motor position for motor $i$ at sample $k$, $M_i(K)$ is the measured motor position at discrete time $k$, and $y$ is the measured output at sample $k$ when an estimator and control model is generated. Therefore, if noise or disturbances are detected in the measured output, the model aligns its dynamics using this information with the true model dynamics. On the other hand, if the required model is generated for prediction and simulation, the signal in (22) will be replaced by the previous model estimations at time $k-j\, y(k-j)$. In this case, errors and disturbances detected at the output will not be perceived by the model unless a closed-loop control action modifies the input components of the regressor. In this case, if the model diverges from the real dynamics, they will not align with each other. The model architectures are given in Figure 3.

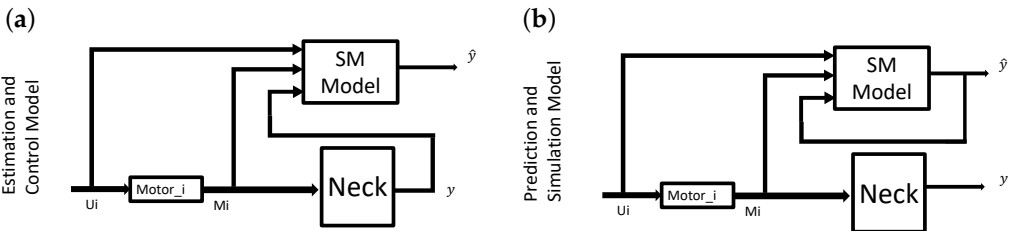

**Figure 3.** Model architectures. (**a**) An estimation and control model. (**b**) A prediction and simulation model.

With the regressor being defined as in (22), we generate our FSS by applying a sum of sinusoidals to each of the three motors such that the signals are not correlated and they give us a wide spectrum of the neck behavior. We capture the real motor positions, desired motor positions and, as output, the neck pitch and roll angles. Then, two separate models are generated.

The signal used for each motor $MP(i)$ to create the FSS is described in Equation (22). The values used for the specific motor are listed in Table 2.

$$MP(i) = (0.6sin(2sin(\omega_1 t + \phi_1) + cos(\omega_1 t + \phi_2))) * \dots$$

$$* \, abs \left[ 3 + sin(\omega 2t) + sin(\omega_3 t + \phi_3) + sin(\omega_1 t - \phi_4) + \dots \right.$$

$$\left. \dots + sin(\omega_4 t) + sin(\omega_5 t + \phi_5) + sin(\omega_6 t + \phi_6) \frac{sin(\omega_7 t + \phi_7)}{2} \right] \quad (22)$$

As it can be seen, the different signals are non-linear and have different frequencies and phases. This provides a pseudo-random interaction and covers a wide range of operational modes of the neck. The frequency spectrum that was chosen is coherent with the system bandwidth, which is 4 (rad/s), generating soft, continuous, and human-like motion in the axes of interest. By modifying the phase and mixing the minimum 0.25 (rad/s) and maximum rad/s frequencies, we aim to explore in a single experiment a wide range of motions providing sufficient information for the FSS. However, it is necessary to point out that some system dynamics did not occur during the proposed study scenario, such as the three tendons pulling at the same time with the same force, which keeps a static neck position with different stiffness, as well as continuous single tendon activation, to name some. Even if the proposed FSS does not cover the full system dynamics, the chosen signal should cover the normal operational range for the soft neck.

**Table 2.** Values used for the identification data set creation.

| Variable | PositionM1 | PositionM2 | PositionM3 |
|---|---|---|---|
| $\omega_1$ rad/s | 1 | 1 | 1 |
| $\phi_1$ rad | 0 | 2.09 | 4.18 |
| $\omega_2$ rad/s | 0.25 | 0.25 | 0.25 |
| $\phi_2$ rad | 0 | 2.09 | 4.18 |
| $\omega_3$ rad/s | 1.5 | 1.5 | 1.5 |
| $\phi_3$ rad | 0.32 | 0.32 | 0.32 |
| $\omega_4$ rad/s | 2.56 | 2.56 | 2.56 |
| $\phi_4$ rad | 0.095 | 0.095 | 0.95 |
| $\omega_5$ rad/s | 1.75 | 1.75 | 1.75 |
| $\phi_5$ rad | 0.09 | 0.09 | 0.09 |
| $\omega_6$ rad/s | 1.66 | 1.66 | 1.66 |
| $\phi_6$ rad | 0.29 | 0.29 | 0.29 |
| $\omega_7$ rad/s | 4 | 4 | 4 |
| $\phi_7$ rad | 0.67 | 0.67 | 0.67 |

The identification and validation data sets are given in Figure 4, where 10,000 samples were taken, 7000 for the FSS (Identification Data Set) and 3000 for the validation set.

As seen in the Figure, the rotational position of the tendon is maximum 6 rad. Having a pulley diameter of 15 mm, each tendon has linear displacement of the tendon 7.5 mm/rad, and therefore, a maximum linear displacement of $\approx$45 mm. This generates inclinations of $\pm20$ deg for the pitch and $[-20, 40]$ deg for the roll. This provides a wide dynamic range of motion. In addition, the motion frequency was set to replicate a human-like motion which is the region of interest.

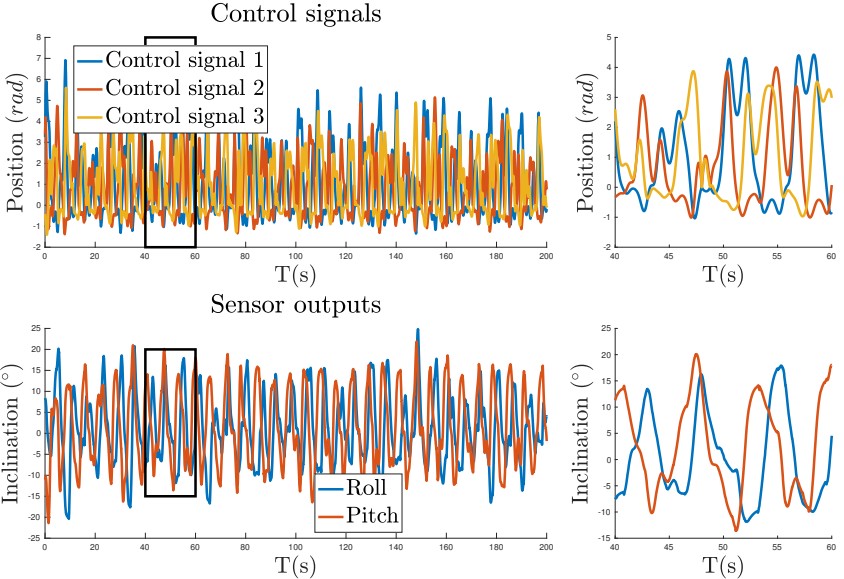

**Figure 4.** Input and output signals.

## 4. Recursive Least-Squares Linear Identification

Despite the nonlinear nature of the plant, a simple recursive least-squares (RLS) identification was performed. On the one hand, it will show a qualitative estimation of the system's nonlinearity degree, and on the other hand, a linear model could be used to solve the nonlinearity issues by means of robust or adaptive control strategies.

The RLS identification algorithm can be described using an ARX structure:

$$\hat{y}(t) = -a_1 y(t-1) - .. - a_{na} y(t-na) + b_1 u(t-1) + .. + b_{nb} u(t-nb), \tag{23}$$

with $y(t)$ and $u(t)$ being the plant output and input variables, with a matrix representation as follows:

$$\hat{y}(t) = \theta \phi'(t-1) \tag{24}$$

$$\theta = [a_1, .., a_{na}, b_1, .., b_{nb}] \tag{25}$$

$$\phi(t-1) = [-y(t-1), .., -y(t-na), u(t-1), .., u(t-nb)] \tag{26}$$

Increasing one time-index ($\hat{y}(t+1) = \theta \phi'(t)$), Equations (24)–(26) provide the output prediction, based on the model parameters ($\theta$), and the set of past inputs and outputs ($\phi(t-1)$). Comparing the next actual system output with this predicted value results in the prediction error:

$$\epsilon(t) = y(t) - \hat{y}(t) \Rightarrow \epsilon(t+1) = y(t+1) - \hat{y}(t+1) \tag{27}$$

In order to minimize this error, different algorithms can be used. In the least-squares case, the squared sum of all errors is the variable to be minimized. Since the parameters that minimize the error produced by the least-squares solution can also be obtained from the preceding parameters (recursively), the algorithm can be expressed using recursion, as shown below:

$$\hat{\theta}(t+1) = \hat{\theta}(t) + F(t+1)\phi(t)\epsilon(t+1) \tag{28}$$

$$F(t+1) = F(t) - \frac{F(t)\phi'(t)\phi(t)F(t)}{1 + \phi(t)F(t)\phi'(t)} \tag{29}$$

$$\epsilon(t+1) = y(t+1) - \hat{\theta}(t)\phi'(t) \tag{30}$$

These equations define the operations needed to find $\hat{\theta}(t+1)$ based on the previous parameters and captured inputs and outputs. An improved method is described in [33] as RLS with a constant forgetting factor (CFF-RLS). It will not be necessary at this point, as the soft neck system identification will be done offline, but it can be used in the future, if an adaptive scheme is proposed as a solution to the nonlinearity issues. See [33] or [36] for a more detailed discussion about RLS and other identification methods.

Using the described inputs and outputs definition of Section 2, the same data captured in the identification experiments were used in order to obtain a plant model. As the data capture is based on the motor positions, we can find the equivalent input values using Equations (9) and (10), and consider these inputs. The outputs will be the same as in the other cases, the neck pitch and roll angles.

For example, Figure 5 shows part of the inputs and outputs considered for the pitch and roll RLS identification.

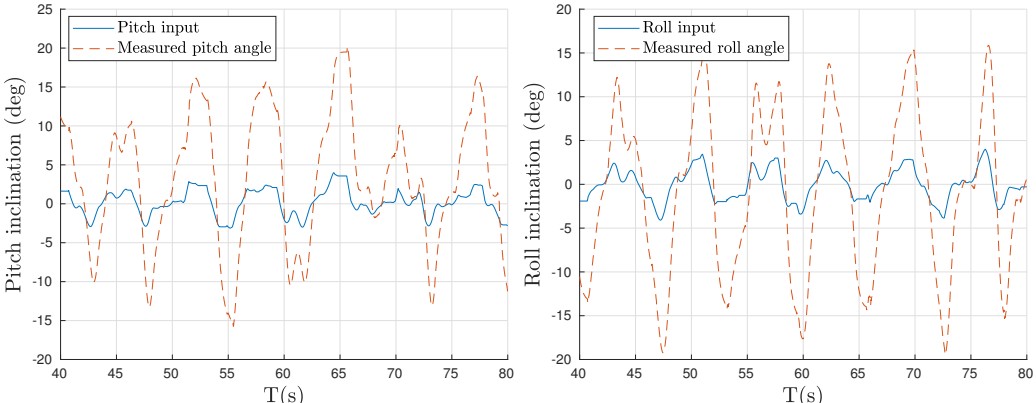

**Figure 5.** Examples of decoupled identification datasets. Redefined input targets compared to measured angles for pitch (**left**), and roll (**right**).

A small delay can be observed in the data set example shown in that Figure. This delay must be considered during the system identification process. The resulting transfer functions corresponding to the pitch dynamics, $G_\theta(s)$, and roll dynamics, $G_\phi(s)$, obtained using the RLS algorithm through the entire data set, are:

$$G_\theta(s) = e^{-0.08s}\frac{27.691}{(s+5.293)} \quad G_\phi(s) = e^{-0.08s}\frac{25.424}{(s+4.938)}. \tag{31}$$

The model unit input time responses are shown in Figure 6 for the described system model. Note how both systems' ($G_\theta(s)$, $G_\phi(s)$) static gains are close to 5, showing a stationary response above the unit input level, as expected from Figure 5.

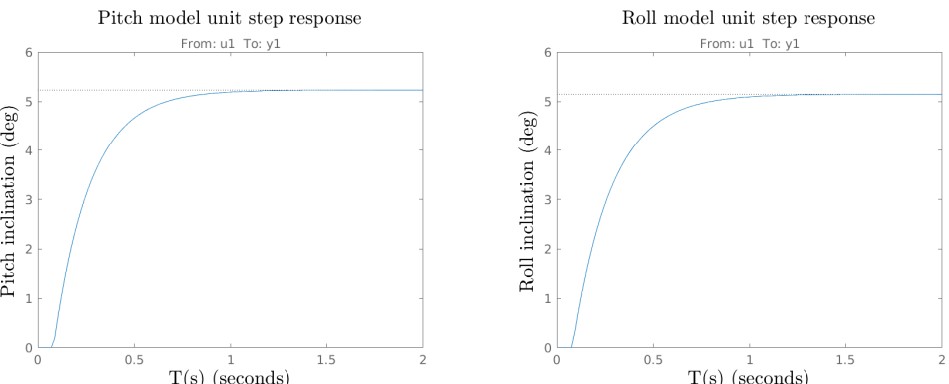

**Figure 6.** Unit input time response for $G_\theta(s)$ (**left**) and $G_\phi(s)$ (**right**).

Once the linear models of the soft neck decoupled system are determined, a new input response simulation was performed using those models, together with a new data set for validation and accuracy check. A partial plot of these results is shown in Figure 7 for the pitch and roll angles.

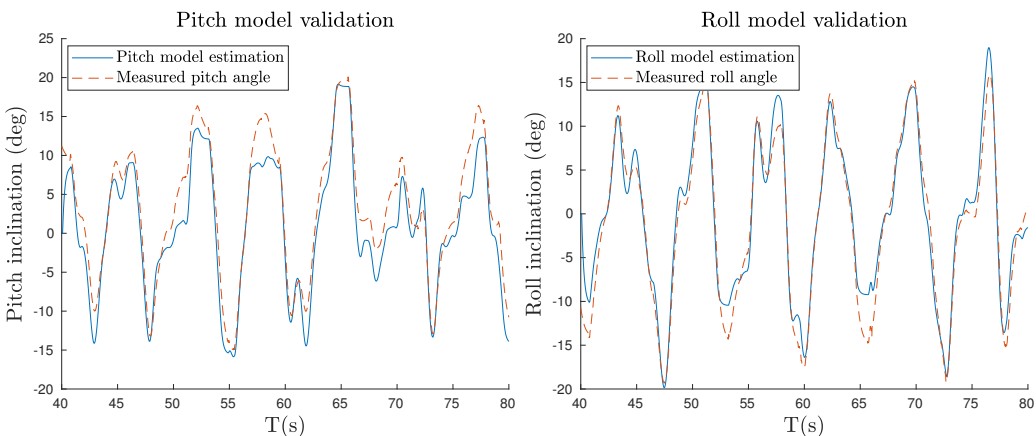

**Figure 7.** Validation example of the identified models for $G_\theta(s)$ (**left**) and $G_\phi(s)$ (**right**).

Note how although the linear model captures the system behavior quite well, there are mismatches due to plant non-linearity. In order to deal with these problems, a robust controller could be used in the future, since it will provide a constant behavior despite plant parameter changes or non-linearities.

## 5. Experimental Setup

The main objective of this work was to extract a 3D model for the soft neck platform. The selected identifications were done in an offline configuration, with an open-loop data capture scheme. In this sense, a set of experiments has been designed for data capture.

As stated before, as we expected a correlation between the motor position and the inclination of the top platform, the motors' states are therefore considered as inputs, while the measured sensor angles are used as system outputs. The captured data involve the following inputs and outputs:

- Motor input position (*rad*)
- Motor current position (*rad*)
- Motor current velocity ($\frac{rad}{s}$)
- Platform roll or model output (º)
- Platform pitch or model output (º)

Neck actuator motion was programmed to follow a composition of sinusoidal functions, as described in Section 3.1. The captured motion describes human-like movement. Input and output sets can be seen in Figure 4.

All models used the same data for modeling the system, enabling direct comparisons of the models. Additional tests were also captured for validation purposes.

## 6. Model Results

This section presents the different model behaviors for the validation data. Figure 8 represents 30% of the data set described in Equation (22). All the results will be compared to those obtained by a NLARX NN with two hidden layers and 25 neurons each. To train the neural network, the 70% of the FSS data set described in Equation (22) was used.

As a form of comparison, the fitting value for the Normalized Root Mean Squared Error (NRMSE) will be taken into consideration. This tool finds the difference between the measured data and the model response as the sum of the squared individual errors throughout the entire signal. Using this method, the large errors will have a bigger quantitative penalization than small errors.

Finally, to validate the results, three independent different tests will be conducted that compare the methodologies used for static movement, dynamic movement, and normal operation mode.

### 6.1. Set Membership

The set membership model was able to follow the system output with high accuracy (see Table 3). However, in some peaks, especially for the pitch output, it failed to reach the maximum value (an error of around 2 degrees) (see Figures 9 and 10). SM scores a high fit for both pitch, 93.9794%, and roll, 96.8854%. The values reached by the model show possible over-fitting to the training data. This means that the identification data set almost explicitly contains the validation set.

### 6.2. Neural Network

The neural network used is an NLARX with two hidden layers of 25 neurons each. The output for the validation can be found in Figures 9 and 10. Like the SM case, the resulting fit value obtained exceeds expectations and some concerns of over-fitting arise. The fitted values for pith and roll are 99.0162% and 99.2027%, respectively (see Table 3). Similarly to the behavior of the SM case, the training data for the neural network covers the validation data with high precision. Therefore, additional tests are required to properly evaluate the model performance.

### 6.3. Recursive Least-Squares

The RLS model proposed in Section 4 was fed with the validation data in order to be compared with both previous models. The model output can be seen in Figures 9 and 10. RLS captured the overall behaviour of the neck within acceptable tolerance. The mismatch observed is attributed to the plant non-linearity. The fit values are 78.0448% for pitch and 82.7217% for roll. An overall comparison of the results can be found in Table 3.

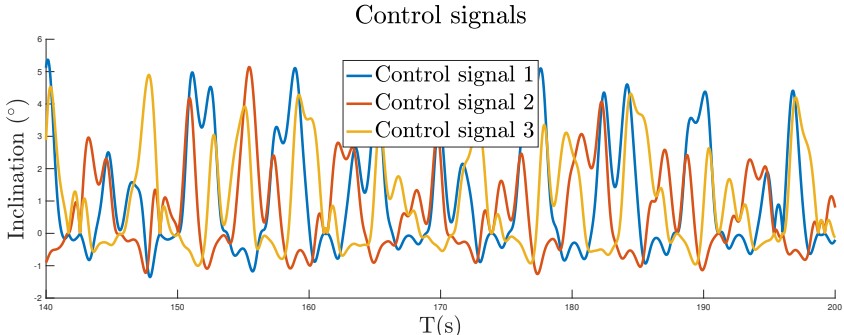

**Figure 8.** Control signal for the validation test.

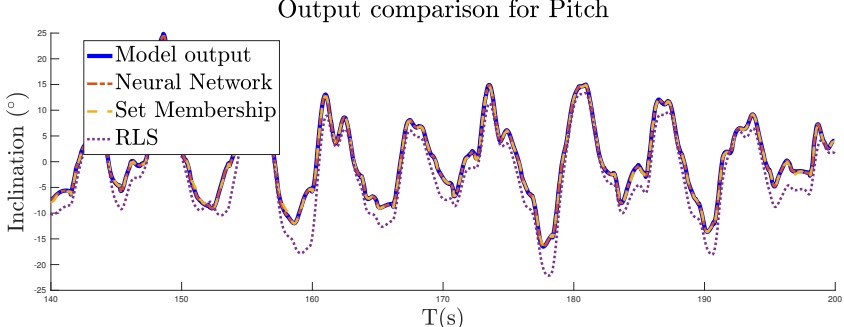

**Figure 9.** Validation of the pitch output.

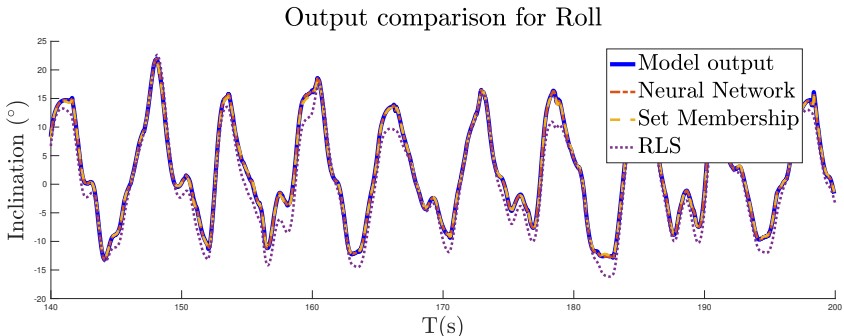

**Figure 10.** Validation of the roll output.

**Table 3.** Fitness of models on the validation data.

|  | Pitch | Roll |
|---|---|---|
| **SM** | 93.9794% | 96.8854% |
| **NN** | 99.0162% | 99.2027% |
| **RLS** | 78.0448% | 82.7217% |

## 7. Methods Comparison

With the models already trained and properly adjusted, three different tests in the estimation configuration were conducted to validate proper behaviour of the systems and compare the outputs for the different models.

### 7.1. Test 1: Step Inputs

This test consists of three separated step-waves with a duration of 8 seconds, each independently activating the neck tendons, as shown in Figure 11. The aim of this test was to validate the models' capability of responding to a static input. Since the dataset for the training and NN which corresponds to the FSS lacks individual tendon actuation, some error in the models is expected. Figures 12 and 13 show the outputs.

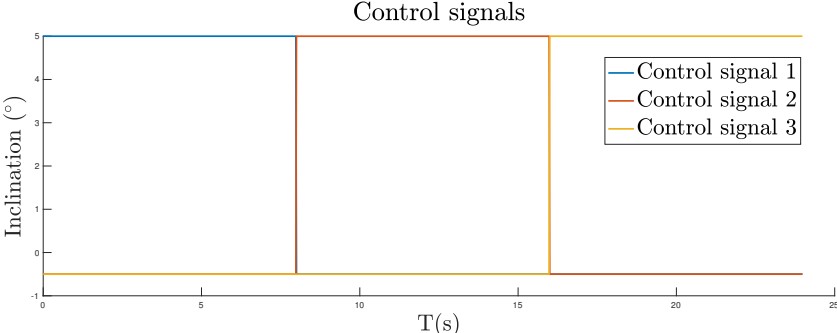

**Figure 11.** Test 1 input signal.

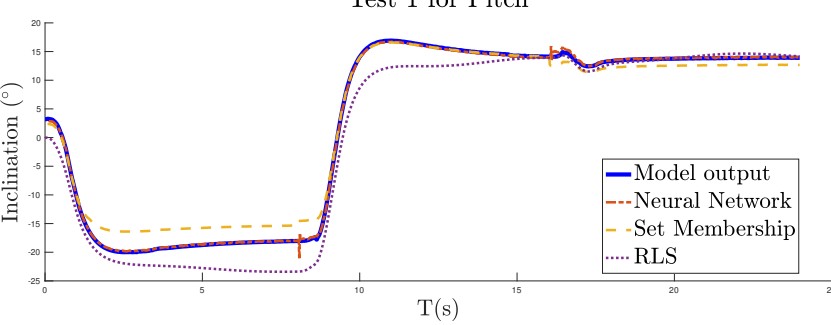

**Figure 12.** Test 1 results for pitch.

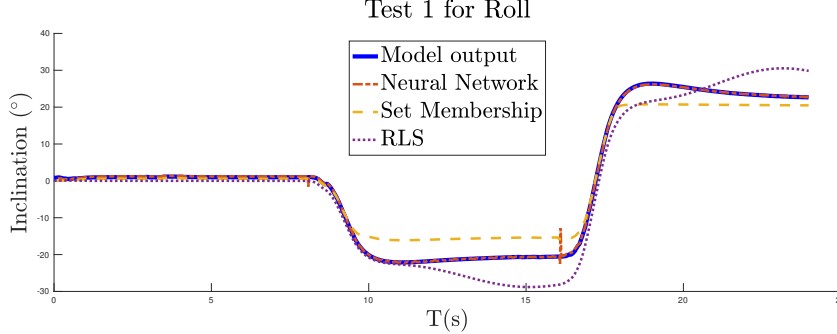

**Figure 13.** Test 1 results for roll.

Set membership scored 87.6260% in the NRMSE test for pitch and 79.3195% for roll. NN followed the output better and scored 97.9574% for pitch and 98.5654% for roll. The SM response did not settle at the appropriated inclination, lagging out before the maximum inclinations were reached. As can be seen, the model does not reach the negative inclinations properly. However, in Figure 12, the dynamics for negative and positive inclinations are followed as desired. It is seen that the existing bias in the time windows [3, 5] s follows the output dynamics. In the roll axis case, the same bias appears with the negative inclination angles, while in the positive case, it stabilizes at time 17 s in the final value. However, it does not properly capture that dynamic, neither. The maximum errors for the NN appear at 8 and 16 seconds, when the signal is changed from one tendon to the next; this is probably due to a lack of information in the data set. Meanwhile, RLS scored 79.9230% in pitch and 76.0284% in roll. This can be due to the fact that the initial conditions do not match the real model ones.

*7.2. Test 2: Sine Inputs*

The next test feeds a more complex signal composed of a sine wave with increasing frequency instead of a step signal, as shown in Figure 14. The test is conducted in order to validate the models for simple dynamic movements. As mentioned before, these cases are not explicitly captured by the FSS or training data. Figures 15 and 16 show the outputs.

The Set Membership followed the output of the system closer than in the previous test. It scored 93.0012% for pitch and 93.8389% for roll, much closer to the validation results. It is important to mention that the SM follows the dynamics and does not have the bias error observed in Test 1. Therefore, the FSS properly captures the continuous dynamic behavior, but it requires additional information to capture static behaviors, as required in Test 1. This is true also for the NN model, which also improved. It scored 98.1058% in pitch and 98.3087% in roll. The RLS roll output shows the disadvantages of this model. Due to the decoupling of the signal, while the initial movement causes little movement on the roll axis, RLS cannot process them and scores a worse fit than in the rest of the test. The fit scores are 78.4813% in pitch and 48.5090% in roll.

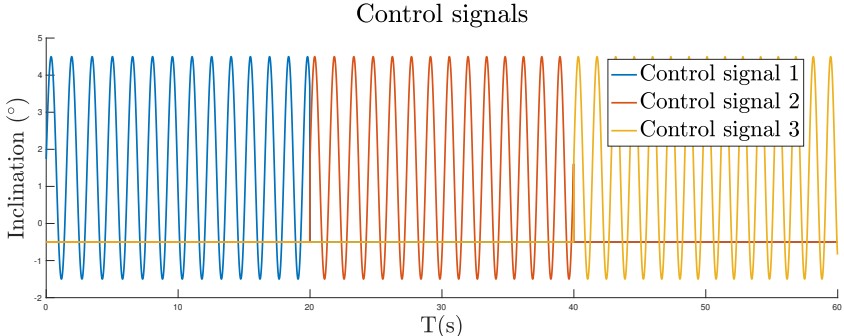

**Figure 14.** Test 2 input signal.

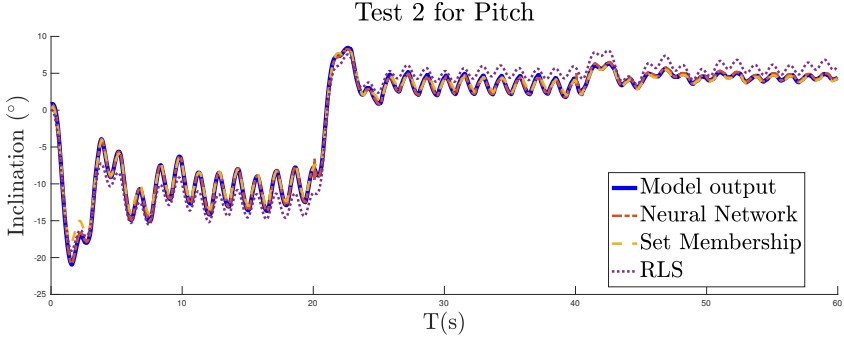

**Figure 15.** Test 2 results for pitch.

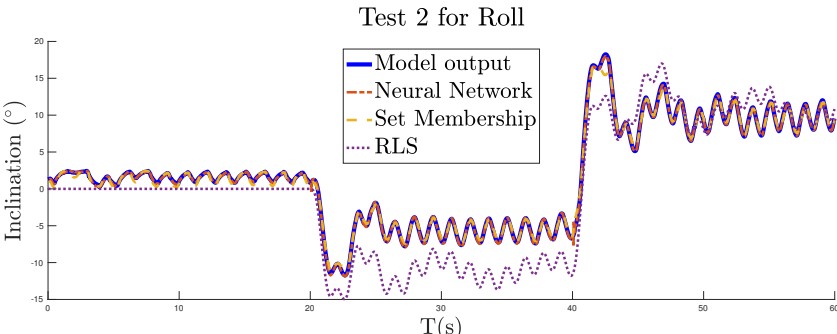

**Figure 16.** Test 2 results for roll.

### 7.3. Test 3: Neck Rotation

The final test combines multiple sine and cosine waves in order to create a rotatory motion on the neck, Figure 17. This test simulates normal operation for the neck, where a circular motion is described. Figures 18 and 19 show the outputs.

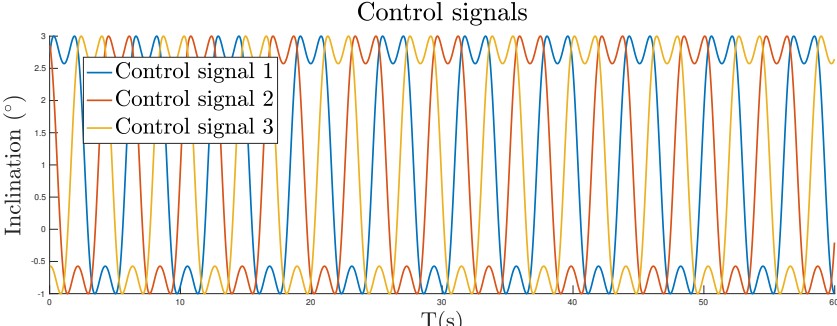

**Figure 17.** Test 3 input signal.

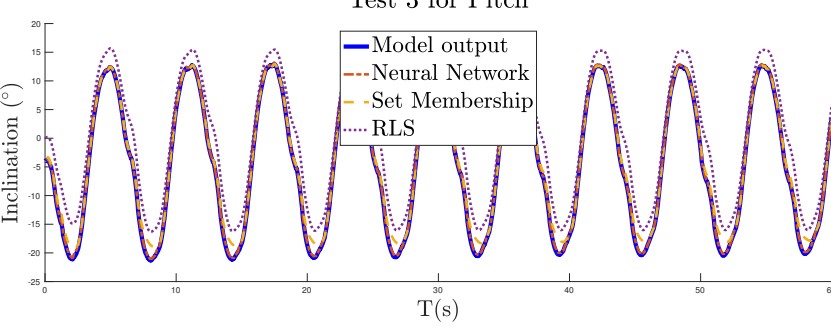

**Figure 18.** Test 3 results for pitch.

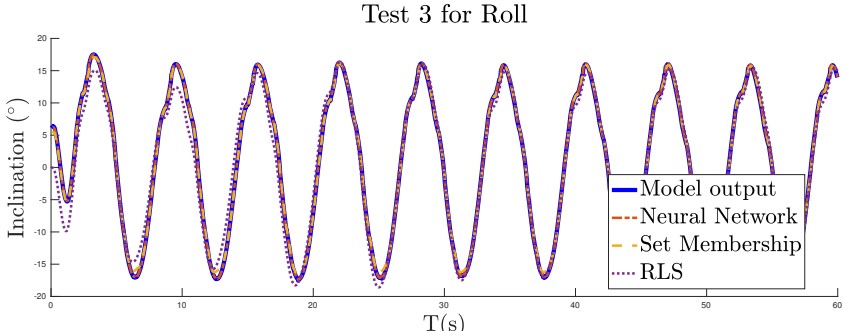

**Figure 19.** Test 3 results for roll.

**Table 4.** Fitness on validation data.

|  | Test 1 | | Test 2 | | Test 3 | |
|---|---|---|---|---|---|---|
|  | **Pitch** | **Roll** | **Pitch** | **Roll** | **Pitch** | **Roll** |
| **SM** | 87.6260% | 79.3195% | 93.0012% | 93.8389% | 90.2614% | 96.7512% |
| **NN** | 97.9574% | 98.5654% | 98.1058% | 98.3087% | 98.5191% | 99.1767% |
| **RLS** | 79.9230% | 76.0284% | 78.4813% | 48.5090% | 63.1412% | 83.2230% |

SM scored 90.2614% on pitch and 96.7512% on roll, while NN scored 98.5191% for pitch and 99.1767% for roll. NN captures the dynamics of the neck both in pitch and roll. SM also closely resembles the output for roll, but on pitch, lags behind the real value. RLS also resembles the real output, but with an offset in pitch. The final scores for RLS are 63.1412% in pitch and 83.2230% in roll.

Table 4 shows a summary of all model scores for all previous tests.

### 7.4. Prediction and Simulation Configuration

In the previous Sections 7.1–7.3, we evaluated the performance for the NN and SM models using the measured output in the model regressor. This configuration can be used for control or estimation applications, to control the plant using the future predicted behaviour. Alternatively, prediction control techniques are desirable. These models are limited to short prediction horizons. On the contrary, if the model is aimed for simulation or pure prediction over long horizons, parallel architecture has to be used as the one in Figure 3. In that case, the model is fed with the previous estimations, and therefore, it can model the whole system's behavior. If the output is disturbed during simulation, the model will not be aligned and will not provide information in this regard.

In order to evaluate the performance of the obtained NN and SM models as predictors, we used Experiments 2 and 3 from Sections 7.2 and 7.3, respectively.

#### 7.4.1. Neck Rotation

When these experiments are applied using the parallel architecture, we can see in Figure 20 corresponding to the pitch that both models decrease in performance. However, the dynamics are still well-captured by both models. In the SM case, the limit values are not reached properly with an error of ≈5 deg for the negative picks and ≈3 deg for positive ones. However, the overall dynamics are captured with a fit that marks 77%. For the NN model, the fit marks 58%. As can be seen, there are important dynamic errors in the negative sinusoidal cycle. Regarding the roll, as shown in Figure 21, both models properly capture the dynamics with fits that mark 86.5% for the SM model and 87.5% for the NN model. RLS results are unchanged, since no feedback is used in the regressor.

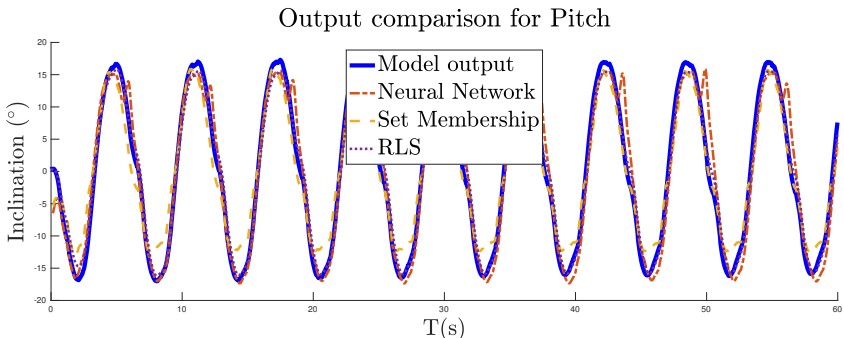

**Figure 20.** Prediction configuration results for pitch in Test 3.

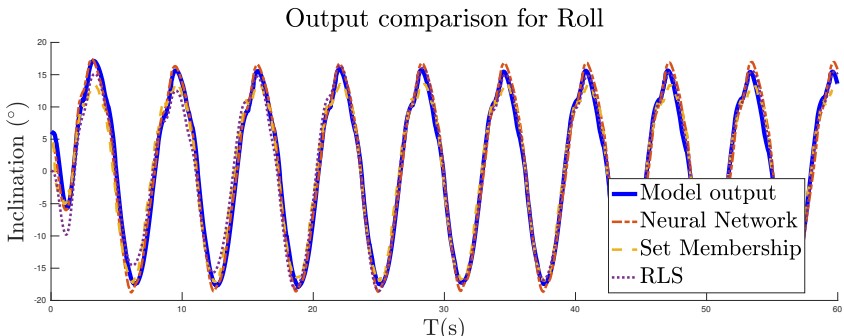

**Figure 21.** Prediction configuration results for roll in Test 3.

### 7.4.2. Sine Wave

Unlike the last results, in Section 7.4.1, as expected, they are not as clean as desired, considering that the FSS lacks these behaviors and therefore, the predictor does not emulate the given dynamics. As can be seen in Figures 22 and 23, the SM stays closer to the measured values. However, there are important gain errors and the model dynamics do not resemble the expected one, even if the results are better than those obtained by the NN model. The final fit values in these cases were: for pitch, NN = 13.5% and SM = 60%; for roll, NN = 60% and SM = 37%. These results confirm that in order to model static and non-coupled behaviours, additional dynamic signals should be considered in the FSS so that the identification data provide reliable information to increase the performance to the one shown in the experiment of Section 7.4.1.

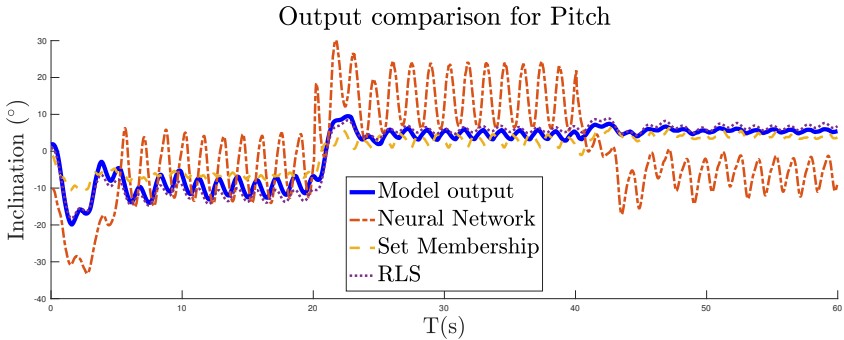

**Figure 22.** Prediction configuration results for pitch in Test 2.

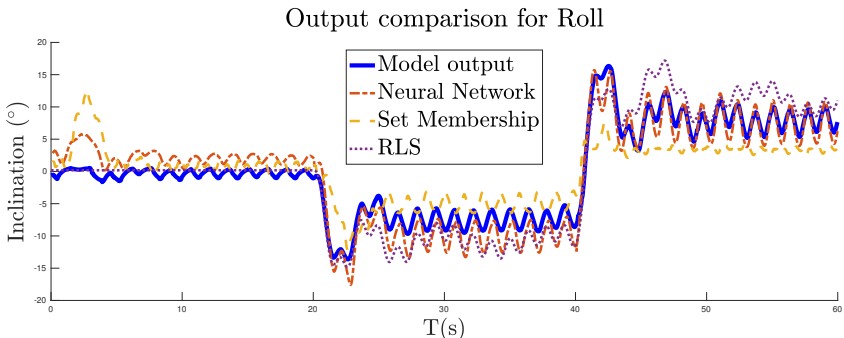

**Figure 23.** Prediction configuration results for roll in Test 2.

## 8. Conclusions

In this work, an improved mathematical model for a robotic soft neck has been presented. The whole soft-neck actuation range was modeled, resulting in a multi-input multi-output (MIMO) system showing a total of three inputs and two outputs. In particular, a nonlinear data-driven identification model using Set Membership, a linear model using Recursive least-squares, and a Neural Network model have been developed and discussed in this paper.

The outstanding results show that the proposed methods are suitable for estimation and control purposes when measures from the output are available to align the models. As shown, given the high level of correlation that the identification data set has over the NN training and the FSS for the set membership, additional identification data are required to use the methods as predictors over long prediction horizons, although results show that the proposed models are viable in soft nonlinear dynamics with multiple inputs and outputs.

A shown advantage of the SM identification stands in the possibility of incorporating additional signal dependency, delays, and unknown dynamics through a richer identification data set which derives from better and more complex modeling without explicit knowledge of the system. Even though the computational time might be a future consideration, there already exist approximation methods to overcome this drawback.

The accuracy difference found between the linear and nonlinear models suggests an important plant non-linearity, as expected. This issue can lead to problems at the time of defining a control strategy, although there are several options which will be explored in upcoming studies.

From the control point of view, the self-aligning characteristic of the given methods provide further knowledge on forecasting in short horizons, which is interesting for predictive and robust control techniques. Besides, the linear model accuracy is good enough to propose solutions like adaptive or robust control, which can provide excellent results. The predictive models' performance shown allows the use of the system for some applications. However, it is limited to continuous mode operation, which yet limits its utility. To overcome this issue, a more informative data set should be constructed that contains additional system behaviors to the continuous operation mode.

**Author Contributions:** Conceptualization, F.Q., J.M., J.A.C.P. and C.A.M.; methodology, F.Q., J.M., J.A.C.P. and C.A.M.; software, F.Q., J.M. and J.A.C.P.; validation, F.Q., J.M. and J.A.C.P.; formal analysis, F.Q., J.M. and J.A.C.P.; investigation, F.Q., J.M., J.A.C.P. and C.A.M.; resources, C.A.M.; data curation, F.Q., J.M. and J.A.C.P.; writing—original draft preparation, F.Q., J.M. and J.A.C.P.; writing—review and editing, F.Q., J.M., J.A.C.P. and C.A.M.; visualization, F.Q., J.M. and J.A.C.P.; supervision, C.A.M.; project administration, C.A.M.; funding acquisition, C.A.M. All authors have read and agreed to the published version of the manuscript.

**Funding:** The research leading to these results has received funding from the project Desarrollo de articulaciones blandas para aplicaciones robóticas, with reference IND2020/IND-1739, funded by the Comunidad Autónoma de Madrid (CAM) (Department of Education and Research), and from RoboCity2030-DIH-CM, Madrid Robotics Digital Innovation Hub (Robótica aplicada a la mejora de la

calidad de vida de los ciudadanos, FaseIV; S2018/NMT-4331), funded by "Programas de Actividades I+D en la Comunidad de Madrid" and cofunded by Structural Funds of the EU.

**Institutional Review Board Statement:** Not applicable.

**Informed Consent Statement:** Not applicable.

**Data Availability Statement:** Not applicable.

**Conflicts of Interest:** The authors declare no conflict of interest.

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
