# Peer review of "3D Model Identification of a Soft Robotic Neck"

_mathematics, doi:10.3390/math9141652_

Round 1

Reviewer 1 Report

This manuscript is about exploring two different identification methods for soft robotic neck 3D dynamics. Observations:

- Abstract is shallow to concisely describe the research efforts.

-- Introduction provides some useful info but seems to be very much insufficient though. Instead of briefly summarizing many previous studies, relevance and critical evaluations should be developed for each reference. Also, the modeling perspectives are poor in terms of both content and format considerations. Cohesiveness and coherence of the section is very much questionable, e.g., the paragraph for lines 30-33. Many broken (throughout the presentation) or one sentence paragraph, e.g., lines 84-86, further adds to the presentation efficacy.

- Indicate the corresponding size information in Fig.1.

- Put the info on lines 112-116 in a table.

-- Decomposing an apparently nonlinear neck operations on three different axes to obtain SISO systems (for identifications) seems to be a mathematical convenience but they are not convincingly justified in the presentation. Line-146 statement is too broad for different levels of neck stiffness. Instead of simply saying "too small" to neglect coupling effects, possible constraint(s) for the soft neck (potentially including the projection factor with nonlinear components, non-homogeneous structures, etc.) should be explored to validate the decomposition. Overlooking dynamical complexities/consequences of nonlinear components is very much troubling, e.g., lines 178-184 seem to imply substituting nonlinear relationships with an averaging operator!

- Explain lines 252-255 to see the rationale and steps of the mentioned process.

- There is something wrong in Fig.3 caption.

- Explain/justify in detail the selection/significance of Table 1 data and (22) for the FSS coverage, etc.

- Figure 4 is not illustrative at all in its current format (for the respective identification/validation sets, signal variation natures, etc.).

- Based on Fig. 7, either the underlying nonlinearity is negligible or the nonlinearity is of an additive form, especially for the roll angle estimation. In both cases, the identification for the original nonlinear system becomes questionable.

--+ Although some results are encouraging, their scopes are limited to limited preset signals while additional results are inadequate, implying a need for a stronger theoretical/simulation framework for superior modeling performances.

- Ensure all text-only Conclusion.

- Some unexpected/noticeable presentation issues are observed, e.g., line-118, Equation-1 (and many more) punctuations, line-267, lines 281-284, line-310, lines 321-325, lines 357-363, Table 3 caption location, etc., implying a need for a revision.

Reviewer 2 Report

This manuscript describes a tendon-driven neck robot and presents methods for modeling it. It is a continuation of a series of papers describing the hardware design and simpler models. In that sense, this manuscript is not as novel as it could be, but contains sufficient novelty and additional considerations that warrant its publication as a distinct paper. 

There is a body of literature in modeling and controlling tendon-driven robots that should probably be more thoroughly cited. For example:

“Autonomous Control of a Tendon-driven Robotic Limb with Elastic Elements Reveals that Added Elasticity can Enhance Learning” by Ali Marjaninejad, Jie Tan, Francisco Valero-Cuevas

2020 42nd Annual International Conference of the IEEE Engineering in Medicine & Biology Society (EMBC), Pages 4680-4686

“Reinforcement Learning and Synergistic Control of the ACT Hand” by Eric Rombokas, Mark Malhotra, Evangelos Theodorou, Emo Todorov, Yoky Matsuoka, Mechatronics, IEEE/ASME Transactions on, Volume 18 Issue 2 Pages 569 - 577

Lens, Thomas, and Oskar von Stryk. "Design and dynamics model of a lightweight series elastic tendon-driven robot arm." 2013 IEEE International Conference on Robotics and Automation. IEEE, 2013.

Motor commands built from a sum-of-sinusoids are used to generate the FSS in Section 3. Table 1 presents them in terms of rad/s, but it is a little unclear to the reader what the practical importance is. In other words, it is difficult to tell how dynamic the resultant movements were - was the neck jerking around like a crazy maniac? Or just rolling around at reasonable speeds. This is important to understand to know what kind of dynamic effects may be in play - tension in the tendons, damping and friction in the soft spine, dynamics or backlash from the gearboxes, etc. From figure 4, the reader can see that the movements take place over hundreds of seconds, but it’s hard to visualize. A short qualitative description in that section of how those commands played out on the robot would be helpful. They seem adequate to activate the nonlinear interactions of the system, but not necessarily all of the nonlinear dynamics that appear under fast movement. To some degree, the step response experiment addresses this. I suggest that this be briefly considered in the introduction - the authors could make the case that the step response experiment is intended to demonstrate what kind of biologically important movements, like the neck movements used by humans and animals, could be performed by this system. 

Line 359: the “fit” is the key outcome variable presented here, and it should be a little more thoroughly described and justified. 

A note about using the neural network to compare: it is good practice to perform a hyperparameter search of some kind when using neural networks. In this manuscript, a particular choice of architecture - activation function, number of neurons, number of layers , etc. - is presented, but it’s impossible to understand whether that performance is representative of NN in general. I think the result stands without it, but it would be much stronger if a NN hyperparameter search were performed. 

Ultimately this is an interesting manuscript, the methods are well described and appear sound, and the result, though modest, is notable and interesting to others.

Reviewer 3 Report

The paper deals with a performance comparative study of three data driven identification models using Set Membership, a linear model using Recursive Least Squares, and a Neural Network proposed approaches, applied for highly robotic soft neck modeled as a multi-input multi-output system with three inputs and two outputs. The proposed models have been properly trained and adjusted and three different tests were conducted; the obtained results show clearly the advantages and the drawbacks of these methods.

The following issues are recommended to improve the paper:

  1. Abstract: the paper novelty is not clearly issued from the Abstract. Please, give a more clear picture on the paper aim and be more specific on the novelty of the “modeling approaches proposed” (or just “selected”?), what type of identification (which parameters) has been done, etc.
  2. Line 100: “therefore, the output considered can be either.” Unclear statement
  3. Line 267: “Once the regressor is defined.” Unclear statement
  4. Recommendation to improve the English style (in a very few cases).
